# Transport of bound quasiparticle states in a two-dimensional boundary superfluid

Samuli Autti [1] ✉, Richard P. Haley [1], Asher Jennings[1,3], George R. Pickett [1], Malcolm Poole[1], Roch Schanen[1], Arkady A. Soldatov [2], Viktor Tsepelin [1], Jakub Vonka [1,4], Vladislav V. Zavjalov[1] & Dmitry E. Zmeev [1]

The B phase of superfluid $^3$He can be cooled into the pure superfluid regime, where the thermal quasiparticle density is negligible. The bulk superfluid is surrounded by a quantum well at the boundaries of the container, confining a sea of quasiparticles with energies below that of those in the bulk. We can create a non-equilibrium distribution of these states within the quantum well and observe the dynamics of their motion indirectly. Here we show that the induced quasiparticle currents flow diffusively in the two-dimensional system. Combining this with a direct measurement of energy conservation, we conclude that the bulk superfluid $^3$He is effectively surrounded by an independent two-dimensional superfluid, which is isolated from the bulk superfluid but which readily interacts with mechanical probes. Our work shows that this two-dimensional quantum condensate and the dynamics of the surface bound states are experimentally accessible, opening the possibility of engineering two-dimensional quantum condensates of arbitrary topology.

At the lowest temperatures (and here in zero field and pressure) bulk superfluid $^3$He exists in the B phase, formed of triplet-paired Cooper pairs. Here, the minimum energy required to create a quasiparticle, or one half of a broken Cooper pair, is the energy gap $\Delta \approx 1.6$ mK ($2 \cdot 10^{-26}$ J). These quasiparticles are responsible for the macroscopic transfer of momentum and energy in the superfluid, and the quasiparticle density decreases exponentially with decreasing temperature. Therefore, at temperatures below a quarter of the superfluid transition temperature $\approx 1$ mK, the bulk superfluid only conducts heat efficiently from sources hot enough to create new quasiparticles. However, within roughly a coherence length $\xi \approx 80$ nm of the sample container walls, the energy gap is partially suppressed[1–3], as shown schematically in Fig. 1a. The coherence length is the smallest length scale across which the superfluid wave function can change, and therefore the gap suppression region is effectively two-dimensional. This suppression gives rise to a quantum well allowing quasiparticles to exist at arbitrarily small energies[4–13].

In the simplest description, the bound quasiparticles have a linear dispersion as a function of their in-plane momentum $p_{||}$, $E = v_L p_{||}$[5–10,14].

Here $v_L = 27$ mm s$^{-1}$ is the Landau critical velocity, which means that such bound quasiparticles move at a uniform group velocity $v_{qp} = v_L$ in the plane of the surface[7,15]. Since this dispersion is best approximated when the scattering from the containing wall is (partially) specular[7], the primary experiments in this article were carried out with approximately two monolayers of solid $^4$He coating all surfaces[4], yielding specularity in the range between 0.2 and 0.8, where one is full specularity and zero means diffuse surface scattering. We can also measure the effect of removing the $^4$He coating[16] so that specularity is approximately zero. In this case, the bound quasiparticles have a more complicated dispersion, but the order of magnitude of $v_{qp}$ remains the same[7,15].

We are able to expel some bound quasiparticles to the bulk superfluid when a probe inserted in the bulk superfluid is accelerated to a velocity exceeding a critical velocity. The probe we use is the crossbar of a cylindrical goalpost-shaped wire[17] (radius $R = 63.5$ μm, crossbar 8 mm, legs 25 mm, see Methods). When the wire is moved, driven by the Laplace force in a magnetic field, the superfluid flow velocity with respect to the crossbar surface follows $v_{fl} = 2v \cos(\theta)$,

[1]Department of Physics, Lancaster University, Lancaster LA1 4YB, UK. [2]P.L. Kapitza Institute for Physical Problems of RAS, 119334 Moscow, Russia. [3]Present address: RIKEN Center for Quantum Computing, RIKEN, Wako 351-0198, Japan. [4]Present address: Paul Scherrer Institute, Forschungsstrasse 111, 5232 Villigen PSI, Switzerland. ✉e-mail: s.autti@lancaster.ac.uk

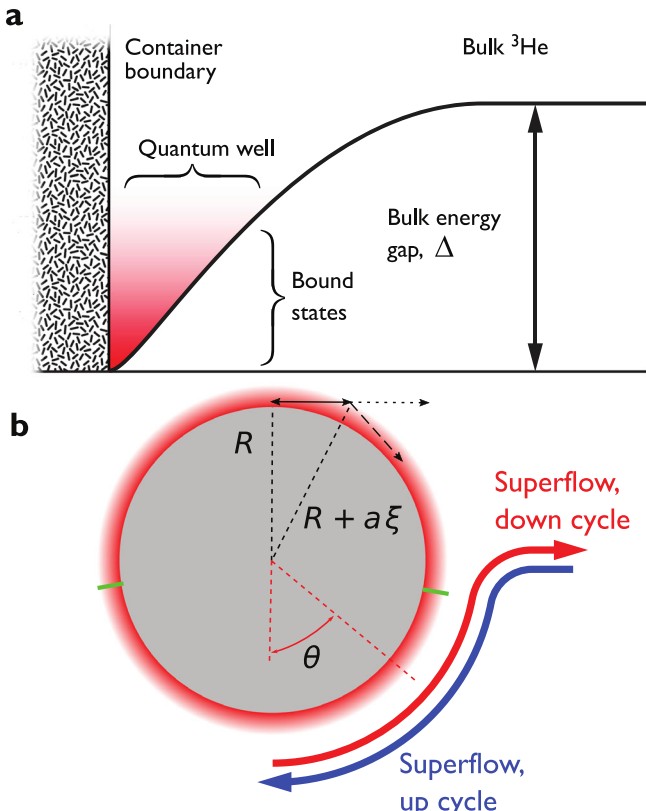

**a** Container boundary

Bulk $^3$He

Quantum well

Bulk energy gap, $\Delta$

Bound states

**b**

$R$

$R + a\xi$

$\theta$

Superflow, down cycle

Superflow, up cycle

**Fig. 1 | The two-dimensional quasiparticle quantum well. a** The component of the superfluid gap that corresponds to momenta perpendicular to the wall is suppressed as the container boundary is approached. This yields a potential well in which bound quasiparticles (red halo) can exist to arbitrary low energies. The gap for motion along the surface remains nonzero. At low temperatures, the density of quasiparticles in the bulk superfluid is vanishingly low. **b** The probe wire (cross-section shown by the grey disk) is surrounded by the bound quasiparticle potential well (red halo). The thickness of the potential well $a\xi$ ($a$ ~ 1, see Methods) determines the bound quasiparticle mean free path as indicated by the double arrow. Here $R$ is the radius of the wire crossbar. When the crossbar is moving, the local superflow velocity around it depends on the polar angle $\theta$. Green notches indicate the span of $\theta = \pm 80°$ where the quasiparticle escape condition $v_\mathrm{fl} > v_\mathrm{c}$ is satisfied locally for the wire velocity of $v = 45 \text{ mm s}^{-1}$.

where $v$ is the velocity of the crossbar in the laboratory frame, and the polar angle $\theta$ is defined in Fig. 1b. The bound-state dispersion is Doppler shifted in energy by $\pm v_\mathrm{fl} p_\mathrm{F}$ as shown in Fig. 2 and detailed in Supplementary Fig. S1 ($p_\mathrm{F}$ being the Fermi momentum), and in the wire frame the bulk superfluid is moving at $v_\mathrm{fl}$. As the energy difference between the highest quasiparticle states occupied at zero temperature and the bulk superfluid is $\Delta$ in the absence of flow, the critical velocity where the first bound quasiparticles can escape to bulk is $v_\mathrm{c} = \Delta/(3p_\mathrm{F}) = v_\mathrm{L}/3$ (see detailed derivation in Methods). Below the critical velocity, we can manipulate the bound quasiparticle dispersion curves by changing the direction and amplitude of the superflow along the surface, allowing us to create a non-equilibrium state within the two-dimensional superfluid. The crossbar trajectories used for this purpose are illustrated in Fig. 3. We return to the details of these patterns of motion in the next section.

As the wire velocity $v$ is increased from zero beyond $v_\mathrm{c}$, the most energetic bound quasiparticles escape to the bulk superfluid as schematically shown in Fig. 2. Details of the escape process and the critical velocity are provided in Methods. While the wire is moving at a uniform velocity[18], a new equilibrium of bound quasiparticle distribution is established, and no further quasiparticles are released into the bulk. A similar process takes place during the deceleration phase at the end

of the motion, where a further pulse of bound quasiparticles can escape. These steps form the first part of the double cycle (green line in Fig. 3).

The pulses of quasiparticles released into the bulk increase the temperature of the bulk liquid. We can infer the dependence between the temperature increase and the heat released by the motion of the wire. Moreover, because each quasiparticle in the bulk liquid carries energy very close to $\Delta$, this measured temperature rise yields the number of quasiparticles that have escaped during the motion of the wire. We label the direction of the initial acceleration-deceleration cycle "up". The quasiparticle release process in an up cycle is schematically illustrated in Fig. 4a.

We know that bulk superfluid flow can be used to push the bound quasiparticles to the bulk superfluid, where they become regular thermal quasiparticles. Simultaneously, the distribution of the quasiparticles that remain bound to the surface is distorted. The basic physics of the bound states that did not escape to the bulk, such as transport within the quantum well, remain unexplored. Thus, it has not been clear whether the two-dimensional system is an independent superfluid condensate with meaningful transport physics of its own excitations, a layer of normal fluid, or merely the border of the bulk superfluid with only local dynamics.

In this article, we use the bulk escape process to take snapshots of the dynamics of the quasiparticles that remain bound to the goalpost wire crossbar. We argue that the bound quasiparticles are decoupled both from phonons in the container wall (crossbar) and from thermal quasiparticles in the bulk superfluid, substantiated by direct measurement of energy conservation in the bound state system. Our experiment also shows that instead of interacting with the bulk superfluid, the bound quasiparticles have their own mode of transport by diffusion within the two-dimensional superfluid. Finally, the observed characteristic time scale of the transport is consistent with bound states in a two-dimensional region of superfluid, not normal fluid. Combining these observations, we conclude that the surface forms a two-dimensional superfluid, separated from the bulk by its different superfluid gap spectrum. This two-dimensional superfluid provides the primary system at low energies that a mechanical probe immersed in the superfluid interacts with.

## Results

We can measure and analyse the heat released by an up cycle of the crossbar by varying the bulk quasiparticle density. That is, the crossbar also scatters bulk quasiparticles while it moves, with the resulting drag force $F$ yielding the heating $Q = Fl$ ($l$ is the distance travelled). This drag force can be varied by changing the temperature of the bulk superfluid and measured independently using a separate thermometer wire whose resonance width $\Delta f \propto F$. The proportionality constant is calculated in Methods, yielding the black line in Fig. 5a, in good agreement with the measured $Q$. The effect of $Q$ can thus be removed by extrapolating the measured heat release linearly to $\Delta f = 0$. The heat released from the bound quasiparticles extracted this way from a single acceleration is $q = 12 \pm 3$ pJ, which is in good agreement with the theoretical estimate ~10 pJ as detailed in Methods.

Importantly for the current experiment, at the end of the up cycle (Fig. 3), the dispersion curves of the quasiparticles have returned to their zero-velocity profile, but leaving on one branch a deficit where the highest energy quasiparticles have energies well below zero (Fig. 4a). Since at low temperatures the equilibrium density of quasiparticles in the bulk liquid is very low, the mechanism for the deficit of quasiparticles to be replaced by quasiparticles coming from the bulk is too slow to be directly observed. We can estimate the time constant of this process from the thermalisation time of the bulk superfluid, ~1 s, and the ratio of the surface areas of the entire container, including heat exchangers and the goalpost wire crossbar, yielding $\tau$ ~ $10^3$ s (details can be found in Methods). Therefore, equilibrium for these particles

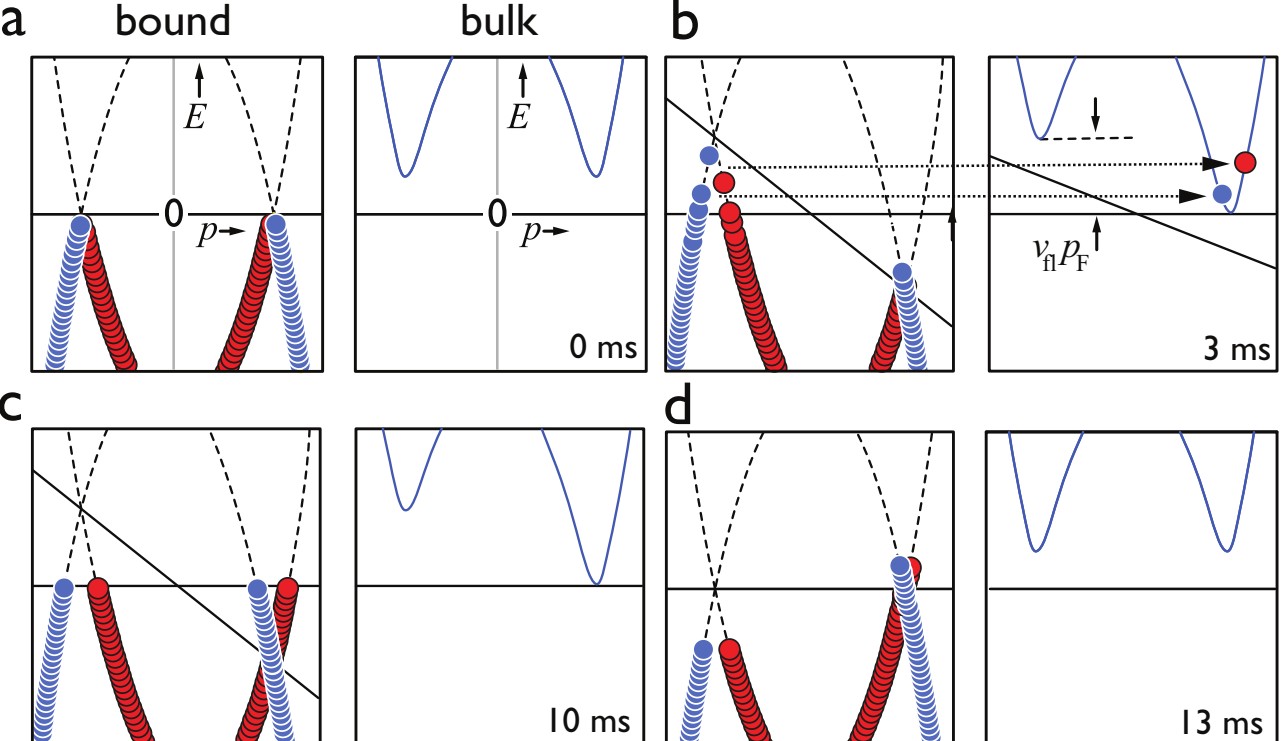

**Fig. 2 | Schematic presentation of the quasiparticle dispersion curves.** The panels represent dispersion curves for the bound quasiparticles (red and blue discs) and for quasiparticles in the bulk superfluid during an acceleration, steady velocity, and deceleration cycle with the indicated times referring to the initial "up" sequence shown in Fig. 3. The dispersion curves are drawn in the reference frame moving with the wire. In panel **a**, the wire is stationary. As we apply an increasing superflow along the crossbar surface during the first 3 ms, the quasiparticle bands become tipped until, as in (**b**), those bound quasiparticles with energies above the minimum in the bulk liquid escape into the bulk. When the acceleration ceases, no more bound quasiparticles can escape, and equilibrium in the wireframe, as shown in panel (**c**), is re-established by diffusion as the wire is moving at constant velocity. On deceleration, a second burst of bound quasiparticles escapes into the bulk. Finally, with the superflow velocity again zero, as shown in panel (**d**), the bands return to their initial state, leaving the bound quasiparticles to redistribute via diffusion. A more detailed description of the same process is shown in Supplementary Fig. S1. Note that this figure is for illustration purposes only. The local dispersion curves cannot be represented in this way, especially for the depletion situation, but it gives the gist of the idea.

can be reestablished only by the flow of bound quasiparticles along the potential well around the crossbar, removing the momentum imbalance, and by flow along the legs of the goalpost wire and container walls, removing excess energy.

Now we can progress to measuring bound quasiparticle transport in the surface system. Referring back to the cycle of crossbar motion shown in Fig. 4a, the end state is not the same as the initial state, as we are left with a quasiparticle deficit in the left-hand branch. The key concept of the experiment is to repeat the acceleration–deceleration cycle once more after a wait of $\Delta t$ (up–up cycle), as shown in Fig. 3. This can be done for two extrema. If we repeat the cycle immediately after the first has ended, that is, setting $\Delta t$ to zero, then the second cycle follows that shown in Fig. 4b, where we start already with the full deficit in the left-hand branch. However, this second process only yields one burst of quasiparticles that on the deceleration. Thus, the combined series of two cycles yields three bursts of quasiparticles into the bulk. Alternatively, after the first cycle, we can wait for an "infinite" period, i.e. longer than the time taken for the deficit to fill by the flow of localised quasiparticles around the wire periphery, in which case we have the same starting conditions as shown in Fig. 4a. Thus, a series of two cycles with a long wait in between will emit four equal bursts of quasiparticles. Between these two extremes, we can explore the situation with intermediate values of the delay $\Delta t$ and can map out how rapidly the deficit fills. That is the basis of the measurement, which provides a measure of the diffusion rate of the localised quasiparticles through the potential well around the wire.

## Bound state diffusion

To interpret the data, we now need to devise a model for diffusive quasiparticle transport in the two-dimensional surface quantum well, which we do following the theoretical lead of refs. 19–22. This model is based on three assumptions (all well justified). First, there is no interaction between surface-bound and bulk quasiparticles, as has been observed across two orders of magnitude in bulk quasiparticle density[4]. This assumption is further substantiated by an estimate of the equilibration time (in Methods) that implies that direct equilibration between the two systems would take at least $10^3$ s. Secondly, the thermal Kapitza resistance between phonons in the solid boundary material and the superfluid quasiparticles is very large at the lowest temperatures. A simple estimate for the exponential decay of the energy of the bound quasiparticles into the material of the crossbar yields time constant $\tau_{R_K} > 10$ s (Methods). These two assumptions imply that energy in the bound quasiparticle system is conserved. Third, the distribution of the bound quasiparticle system reflects the history of crossbar motion as detailed below and in ref. 4, meaning that the bound quasiparticle momentum distribution equilibrates with the time constant $\tau$. Thus, quasiparticle transport acts as the equilibration mechanism.

The diffusion coefficient for quasiparticle transport can be estimated as $D \sim v_{qp} l_{\parallel}$. Here $l_{\parallel}$ is the mean free path which, in the absence of quasiparticle–quasiparticle collisions (this assumption is confirmed below), is determined by the thickness of the quantum well, that is, the distance between the wire surface and the edge of the surface layer,

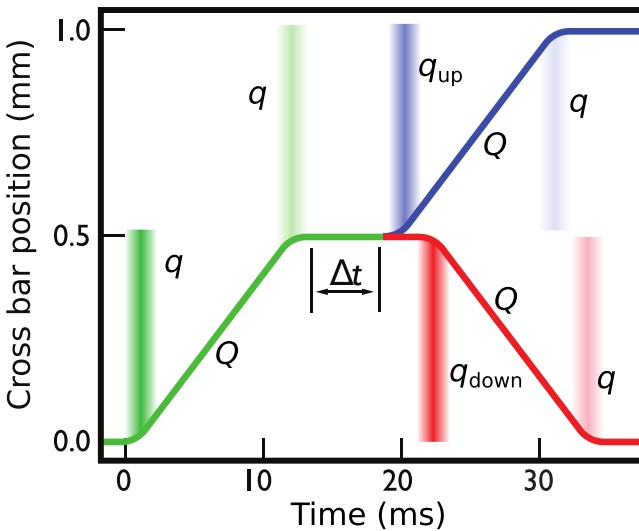

**Fig. 3 | Schematic illustration of the crossbar motion.** The wire can be moved across 0.5 mm with a steady velocity, here 45 mm s$^{-1}$, and then paused (green line). After a wait of $\Delta t$, the process can be repeated with a further up movement ("up cycle" blue line) or reversed back to the starting point ("down cycle", red line). The combination of the green and blue lines is denoted "up-up cycle" and that of the green and red lines "up-down cycle". The vertical bands of colour indicate where surface quasiparticles are emitted from the wire during acceleration and deceleration. The emitted quasiparticles increase the temperature of the bulk superfluid, which is detected using a separate thermometer.

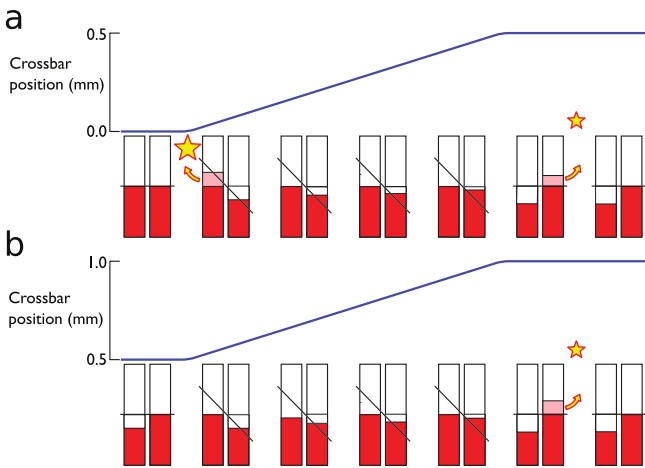

**Fig. 4 | Snapshots of the quasiparticle transport in an up–up cycle. a** As the crossbar starts to move (blue line), the bound quasiparticles in the quantum well (red pillars) are Doppler-shifted up for quasiparticles with momenta along the superfluid and down for momenta against the flow direction. This raises the most energetic bound quasiparticles above the minimum energy in the bulk superfluid, allowing the quasiparticles to escape into the bulk as a sudden burst (yellow star and arrow). During the motion, the quasiparticle deficit created on the right-hand side band can only be filled slowly from the transport of quasiparticles along the potential well. When the motion ceases, the bands return to their original positions allowing another sudden burst of quasiparticles to escape. At the end of the whole velocity cycle, the left-hand side band is left with a deficit. **b** If the initial up cycle is followed with no delay by another up cycle, the second cycle will start with a quasiparticle deficit allowing no quasiparticle emission during the acceleration. Therefore, only one pulse of quasiparticles is emitted from the deceleration. Allowing for (partial) recovery by delaying the second cycle by $\Delta t$ and measuring the amount of emitted quasiparticles thus allows us to take snapshots of the bound quasiparticle diffusion.

$l_{\parallel} \sim 10$ μm (Fig. 1b and Methods). When the crossbar is stopped, the density of bound quasiparticles carrying the momentum imbalance, $n$, reflects the flow velocity along the crossbar surface immediately before the wire is stopped. We estimate this distribution as $n \propto v_{\text{fl}} \propto \cos\theta$. Solving the diffusion equation for this initial state yields exponential recovery of equilibrium $n$ with the time constant $\tau = R^2/D \sim 10$ ms (see Methods). We emphasise that the time constant is tied to the population gradient arising from the flow velocity profile and, hence, to non-local diffusion.

We can probe the diffusive bound-state redistribution by varying the delay $\Delta t$ between the two cycles of motion. Fewer quasiparticles are released in the second cycle if the susceptible population has been depleted during the first cycle. The replenishment of the deficit begins during the deceleration and continues until the velocity reaches full velocity again upon subsequent acceleration. According to the diffusion picture, the energy released in the second acceleration is $q[1 - \exp(-(\Delta t + t_0)/\tau)]$, where $t_0 = 6$ ms is the combined acceleration and deceleration time. The total energy release from the up–up cycle is detailed in Methods. The measured $\Delta t$ dependence, shown in Fig. 6, confirms the exponential equilibration with the fitted value $\tau = 6 \pm 3$ ms[4] in good agreement with the theoretically estimated time constant.

When the first motion cycle depletes the bound quasiparticles available for bulk escape, none will be released if the cycle follows with no intermediate recovery time. That is, the fitted $q$ should be consistent with that extracted by temperature extrapolation above. The blue lines in Fig. 6 fit with $\tau = 6$ ms at two different temperatures, with the average value $q = 11 \pm 5$ pJ. This is in good agreement with $q = 12 \pm 3$ pJ obtained above with the temperature extrapolation. Combining this with the observation detailed below that $q$ does not depend on bulk temperature yields two immediate conclusions. First, starting the second cycle before the bound state recovery is complete provides a quantitative snapshot of bound quasiparticles' dynamics. Second, the bound state population available for bulk escape at this velocity is fully depleted by the first cycle of motion.

We also note that none of the measured dissipation originates from the direct creation of bulk quasiparticles. That is, the temperature extrapolation yields the total magnitude of dissipation at zero bulk temperature, which matches the magnitude of the diffusion process that ties the observed dissipation to the bound state dynamics and, thus, to the bound state escape process. This is despite the fact that a large bulk superfluid volume flows at speeds exceeding the Landau critical velocity when the wire is moved, and we might expect a direct pair-breaking mechanism to arise[23]: In the laboratory frame, the superflow around the crossbar exceeds $v_L$ up to 80 μm above and below the wire vertices. As the crossbar moves across 0.5 mm, the bulk volume where the full Landau velocity is at least momentarily exceeded is, therefore, two orders of magnitude larger than the entire volume of the quantum well around the crossbar that contains the bound quasiparticles. We can speculate that there is no mechanism for the bulk quasiparticles to carry away momentum from the crossbar (unlike the direction-selective escape process of the bound quasiparticles). Thus, bulk quasiparticle creation may be prohibited by the lack of a mechanism to extract energy from the moving crossbar. This observation provides a perspective to experiments carried out in the polar phase of superfluid $^3$He,[24–26] where exceeding the bulk Landau velocity in a large volume potentially causes no observable dissipation either, contrary to theoretical expectation[27,28].

Reversing the direction of motion for the second cycle (up–down cycle, see Fig. 3) results in a temporary excess of bound quasiparticles available for escape during the acceleration of the down cycle. This scenario is illustrated schematically in the Supplementary Fig. S1. The measured data shown in Fig. 6 confirms this excess. Applying the diffusion picture, we expect the excess bound quasiparticle emission to be removed by diffusion as $q_{\text{down}} \exp(-(\Delta t + t_0)/\tau)$, in good agreement

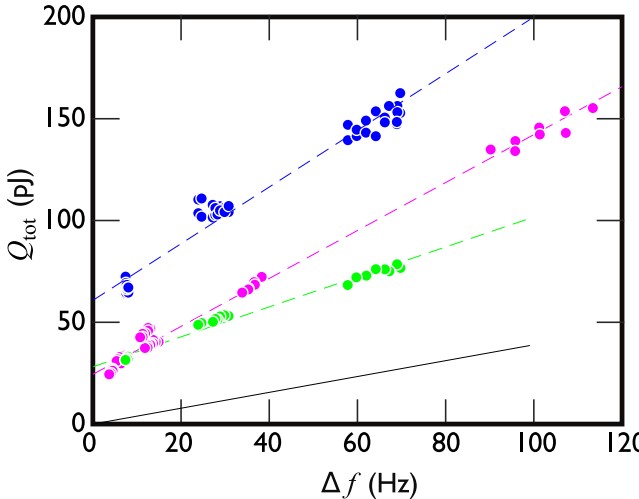

**Fig. 5 | Direct measurement of bound quasiparticle heat release.** The heat $Q_{tot}$ released in an up cycle (green points) depends linearly on the thermal bulk drag force, proportional to the resonance width $\Delta f$ of the vibrating wire thermometer in the same superfluid volume. The dashed lines are linear fits to the measured data. The black line shows an estimate of the heating from the drag force due to thermal bulk quasiparticles (Methods) in good agreement with the slope of the green data considering the estimate is obtained by extrapolating the Andreev reflection force–velocity dependence to four times the velocity where it can be directly measured. Extrapolating the green line to zero $\Delta f = 0$ yields the bound state contribution $q = 14 \pm 3$ pJ (intersection with the y-axis is at $2q$). For an up-up cycle with $\Delta t > 25$ ms (blue points), the slope is doubled because the distance travelled is twice longer, and the $\Delta f = 0$ intersection yields $4q = 60 \pm 4$ pJ. These experiments were carried out with $^4$He preplating. Up-up cycles measured without $^4$He preplating (magenta points) show a 10% reduction in the slope, and the linear fit extrapolates to $4q \approx 28$ pJ.

with the data shown in Fig. 6. The low-temperature fitted $q_{down} \approx 30$ pJ is consistent with a temporary population excess that is removed by diffusion, as detailed in Methods. At higher temperatures, the hysteretic effect explained in ref. 4 distorts the measurement.

### Diffuse boundaries

We can decrease the number of bound quasiparticles susceptible to the bulk flow by removing the $^4$He coating of the surface and thus taking the specularity close to zero[7]. Reducing the surface specularity moves the density of states towards states with lower momenta in the plane of the wire surface. States with zero in-plane momentum gain no energy from $v_{fl}$ and therefore cannot escape to the bulk. Repeating the extrapolation to $\Delta f = 0$ as before yields $q \approx 7$ pJ (Fig. 5). Assuming the escape process is equally effective with and without $^4$He, this implies that the susceptible bound state density is roughly twice higher with $^4$He coating than without it. This is qualitatively in line with the theoretical prediction in ref. 7. The slope of $Q(T)$ is 10% larger with $^4$He coating than without it. This is because surface specularity does not change the bulk quasiparticle density at a given temperature, and the drag force $F$ at large velocities $v$ increases slowly with growing specularity[29–31]. Change in the surface specularity has no effect on the diffusion time constant process within experimental uncertainties[4]. That is, quasiparticle-quasiparticle collisions remain negligible regardless of changes in the density of states, substantiating the assumption that the mean free path is determined geometrically by the thickness of the surface layer and the curvature of the wire crossbar. Note that we can obtain a coarse estimate of the mean free path where limited by quasiparticle-quasiparticle collisions from the known bulk A phase mean free path, which is ~10 mm extrapolated to these temperatures[32].

To add further evidence for the independence of the surface dynamics, we can extract $q$ and $q_{down}$ by varying $\Delta t$ at different bulk temperatures (no $^4$He, Fig. 6b). The bulk quasiparticle density changes by two orders of magnitude over the temperature range studied but, remarkably, the bound state process remains undisturbed, indicated by the constant energy release. This shows that the snapshot

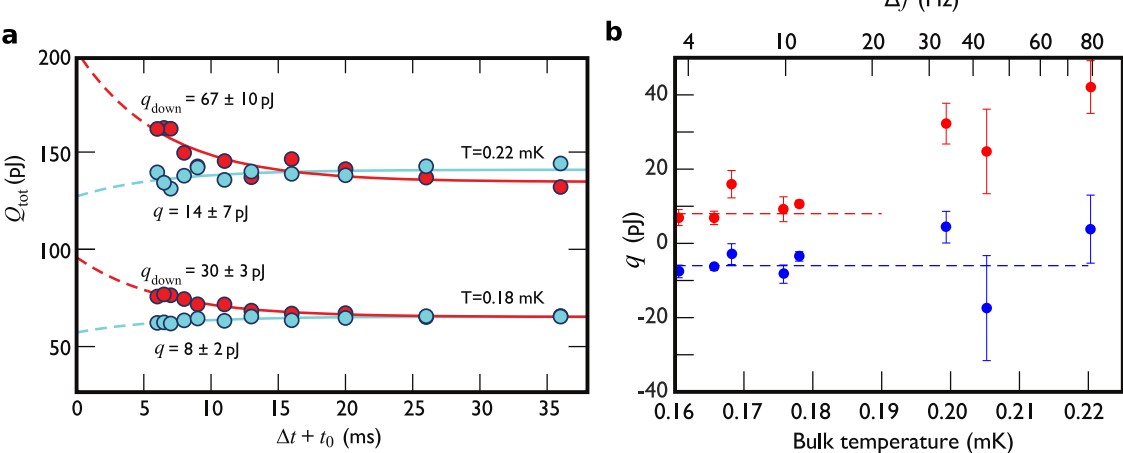

**Fig. 6 | Diffusion in the two-dimensional superfluid. a** We record the total energy released to the bulk superfluid as a function of the recovery time $\Delta t$. Blue circles show the result for an up–up cycle at two different bulk temperatures, and red circles for a corresponding up–down cycle. The fitted exponential time dependencies describe the diffusion process that redistributes quasiparticles until equilibrium is recovered. This allows inferring the magnitude of the bound quasiparticle depletion (or excess for the up–down cycles) that results from the first up cycle. Fitted parameter values $q$ and $q_{down}$ are indicated in the figure with errors corresponding to 68% confidence intervals. Data in this panel was measured with an applied $^4$He coating on the crossbar surface. **b** The number of bound quasiparticles released is approximately halved when the crossbar surface is not coated with solid

$^4$He. We vary the bulk temperature to show that $-q$ (blue points) and $q_{down}$ (red points) are independent of the quasiparticle density in the bulk. The top x-axis shows the thermometer wire resonance width, which is proportional to the density of bulk quasiparticles. Dashed horizontal lines are a guide to the eye corresponding to $-q = -6$ pJ (blue line) and $q_{down} = 9$ pJ (red line). At temperatures higher than 0.19 mK, $Q$ becomes larger than $q$ or $q_{down}$, thus causing difficulties in extracting $q$ and $q_{down}$ reliably. Error bars show the 68% confidence interval of the exponential fits. Additionally, the crossing of the exponential tails seen for the 0.22 mK data in panel **a** is due to a hysteresis effect[4], proportional to $\Delta f$. This acts to increase the apparent $q_{down}$. The magnetic field was 136 mT and $v = 45$ mm/s in both panels.

technique reliably probes the bound quasiparticles' dynamics and that there is no coupling between the bound quasiparticles and thermal bulk quasiparticles.

## Discussion

The phases of superfluid ³He are differentiated by different broken symmetries. Each separate superfluid phase has its own order parameter structure that describes the broken symmetries. The B phase order parameter amplitude (superfluid gap) is uniform in all momentum directions, and thus thermal excitations (normal fluid) vanish exponentially as temperature decreases to zero. In the superfluid A phase, the superfluid gap is zero in one momentum direction, and in the superfluid polar phase, in a plane perpendicular to a specific momentum direction. Thus, in these phases the thermal excitation density does not go to zero exponentially but instead follows a power law[32].

The components of the B phase order parameter amplitude are suppressed within the quantum well near container walls. The gap components are not all uniformly zero in the surface layer, which would be the case for normal fluid[1,6]. For specular boundaries, only the gap component for momenta perpendicular to the wall goes to zero at the wall. For a diffuse boundary, all components are suppressed, but the in-plane components do not go to zero. Thus, the surface region is a superfluid condensate but with a gap spectrum different from the bulk phases[21]. The quasiparticle density in the 2D superfluid can be expected to decrease with a power-law temperature dependence. If we assume the bound quasiparticle density is similar to that in the bulk A phase, their mean free path becomes several millimetres[32], and the thermal quasiparticle population is negligible for the purposes of this Article.

Our experiments show that the 2D layer is characterised by well-defined quasiparticle transport independent of the bulk system. The observed transport time constant is determined by the bound state group velocity -$v_L$, which arises from the gap not being uniformly zero, and the estimated bound state release to the bulk is consistent with the increased density of quasiparticles in the surface layer, arising from the suppressed gap spectrum. The diffusive transport is much faster than the recondensation of the quasiparticles bound to the surface, which is why the added energy is carried away diffusively and re-equilibrated elsewhere on the container walls and the enormous surface area of the sintered heat exchangers.

Heat in this system is contained by the quasiparticles, and thus quasiparticle transport is also heat transport. At energies below the superfluid gap, the surface therefore provides a preferential path for heat flow between hot and cold objects immersed in the superfluid and a direct interaction channel between immersed mechanical probes. These conclusions are supported by anomalous heat transport in superfluid ³He observed independently at Cornell[33], which we suggest results from the flow of heat along the walls of the container and onto the thermometer fork used. In other words, the surface system forms an independent two-dimensional superfluid that, at the zero-temperature limit, determines the thermo-mechanical properties of ³He.

Confining a fermion gas at a low temperature to a high-purity two-dimensional solid-state system has led to the discovery of a variety of quantum Hall phases and topological quantum states. Similarly, the spontaneous formation of ultra-cold superfluid ³He, an extremely pure fermionic system, into a two-dimensional surface system is likely to yield a diversity of physics to be explored. For example, the bound quasiparticles' possible interactions with the sub-gap bosonic excitations or bulk topological defects span at least 7 orders of magnitude in energy below the superfluid gap and some 18 degrees of freedom[32,34]. Our results imply that devising suitable nanoprobes that fit within the two-dimensional superfluid should tap into long-range quasiparticle transport that can be studied in varying topological configurations,

such as across different bulk superfluid phases and interfaces, via controlled confinement provided by engineered nanostructures[33,35–41], or across the free surface[42–47]. It may also be possible to access these phenomena by engineering the topology of the mechanical probes[48,49]. Finally, the surface layer is also expected to host Majorana zero modes[7,50–53] that detailed transport measurements may reveal. These research avenues have the potential to transform our understanding of this versatile macroscopic quantum system.

## Methods

All experimental data and parameter values in this Article are for the saturated vapour pressure, which is vanishingly small at ultra-low temperatures. The superfluid transition temperature at this pressure is $T_c \approx 930\,\mu K$. The superfluid sample is contained in a box made from Stycast paper composite containing sintered silver heat exchangers, surrounded by a guard cell also filled with cold ³He[54]. The motion of the goal-post wire[55] in the cell is illustrated schematically in Fig. 7.

### Thermal drag force acting on the goalpost wire

Changes in the bulk thermal quasiparticle density can be measured using a superconducting NbTi 4.5 μm-thick vibrating wire immersed in the same superfluid volume as the goalpost wire. The resonance width of the thermometer wire follows[29,31,56,57]

$$\Delta f = \frac{F_{4\mu m}}{2v_{4\mu m}}\rho\pi^2, \qquad (1)$$

where the resonance width depends on temperature as $\Delta f \propto \exp(-\Delta/k_B T)$, $F_{4\mu m}$ is the peak drag force acting on the resonator per cross-sectional area of the resonator in the direction of motion, $\Delta$ is the superfluid gap and $k_B$ the Boltzmann constant, and $T$ is the superfluid temperature. The ratio $F/v$ is approximately the same for all probes moving in the superfluid (geometric corrections of the order of one do apply), provided the velocity of the probe is smaller than

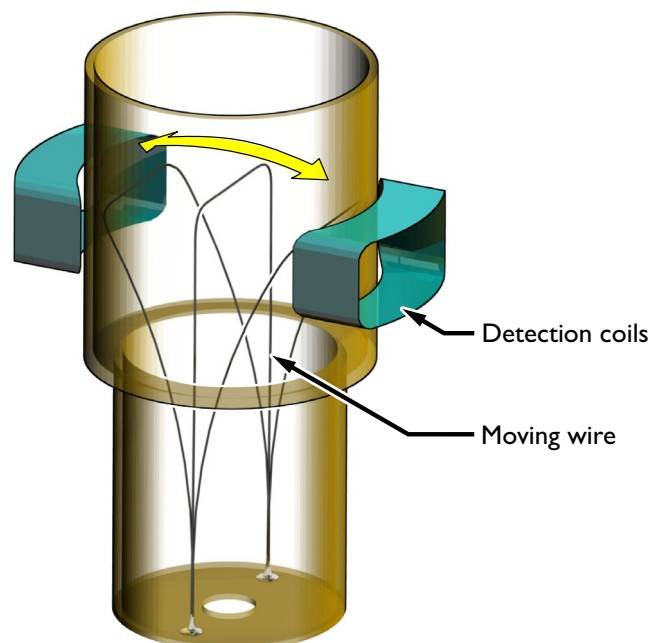

**Fig. 7 | The goal-post shaped wire.** We can move the crossbar of the goal-post wire at a constant speed over a distance of several mm, as indicated by the yellow arrow, enabling the current experiments[18]. The wire is surrounded by a volume of superfluid used as a bolometer for detecting the heat released from the motion of the wire[4]. The detection coils are used for calibrating the velocity of the wire crossbar.

roughly 1 mm s⁻¹. $\Delta f$ is measured by operating the thermometer wire well below this threshold.

At velocities larger than 1 mm s⁻¹, the ratio $F/v$ decreases because of nonlinear effects that arise owing to Andreev reflection of quasiparticles. We estimate this effect for the goalpost wire ($v$ = 45 mm s⁻¹) using Equation 17 in ref. 29 (see also ref. 31). Note that this strictly speaking applies to specular surface scattering only. The diffuse model gives $F/v$ ratios approximately twice smaller at $v$ = 45 mm s⁻¹.

We can use this to estimate the thermal drag force acting on the goalpost-shaped wire. That is, the power dissipated by the thermal drag force acting on the goalpost wire is $Fv$. This implies that for a fixed distance travelled, the total energy dissipated is $Q(T) \propto \Delta f$ with the proportionality constant determined by the ratio of the probe diameters, densities, and the probe velocity as described above. Estimating $Q(T)$ this way yields the black line in Fig. 5, in good agreement with measured $Q(T) + 2q$ (green points and line), considering that no fitting parameters were used, precise surface specularity is not known, and only the drag force experienced by the crossbar is included in the estimate (wire legs are excluded). Note that direct verification of the thermal scattering force in this velocity regime is difficult because of the emission of surface-bound quasiparticles that begins at much lower velocities.

## Vibrating wire thermometry and bolometry

The heat release due to ejected bound quasiparticles is monitored by using the surrounding superfluid volume as a bolometer. Changes in temperature are measured using the thermometer wire resonator. Typical bolometer data curves are shown in Fig. 8. The temperature is stable before the goalpost wire is moved at $t$ = 0, and the peak of the bulk temperature increase occurs well after the wire motion ends. This is because the bolometer readout time, determined by thermometer wire resonance width $\Delta f$ and by the thermalisation time

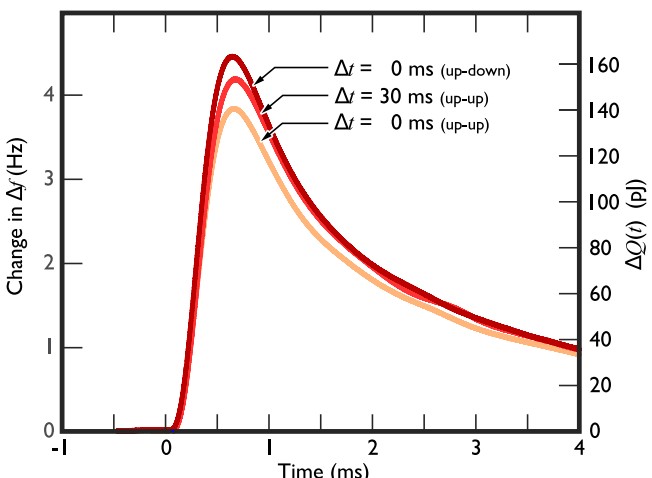

**Fig. 8 | Temperature evolution of the bulk liquid after a double cycle.** The superfluid temperature is stable before the crossbar motion starts at $t$ = 0. The crossbar motion takes less than 30 ms for $\Delta t$ = 0, but the superfluid bolometer reacts slowly to this sudden quasiparticle release. If we look first at the $\Delta t$ = 0 ms up-up data, only three pulses of bound quasiparticles are emitted, producing the smallest rise in temperature (measured as a change in the thermometer resonance width $\Delta f$). For the $\Delta t$ = 30 ms data, four pulses of quasiparticles are emitted, giving a larger temperature swing of the bulk liquid. For comparison, reversal of the direction results in increased (as opposed to depleted) bound quasiparticle release, as shown in the $\Delta t$ = 0 ms up–down data. We can use these curves as transfer functions to infer the level of filling of the left-hand bound state band after the first cycle as a function of the delay time. The right $y$-axis is drawn so that the peak value of each curve corresponds to the total energy released to the bulk liquid. The temperature of the bulk superfluid was 0.22 mK in this measurement. The peaks correspond to roughly 1.7 μK temperature increase in the superfluid.

of the bolometer[58], is of the order of a second. Comparing the two data curves with the delay between the two cycles of crossbar motion equal to 0 and 30 ms, the temperature peak maximum shifts according to delay $\Delta t$ added between the two cycles.

The bolometer is calibrated using resonant AC measurements of the goalpost-shaped wire[59], where the energy output can be directly recorded with a four-point measurement. The data obtained is fitted with the BCS heat capacity[32] using the effective volume of the sample as a fitting parameter. The fitted volume is 16 cm³, which falls between the free volume of the sample container, 15 cm³, and the total volume of the sample container, including the volume within the heat exchangers, 32 cm³.

## Escape condition and the critical velocity

All fermionic quasiparticles in superfluid ³He move at the Fermi velocity ($\approx$50 m s⁻¹). However, the bound quasiparticles are nearly perfectly retroreflected from the edge of the surface quantum well owing to Andreev reflection. If reflection from the wire surface is specular, the resulting bound quasiparticle dispersion is the Dirac dispersion $E = v_L p_\parallel$. The corresponding group velocity $v_L$ is the drift speed arising from the minute misalignment of the inbound and outbound trajectories in the Andreev reflection process[60].

Let us assume that the bound quasiparticle system is in equilibrium at zero temperature. This means that the quasiparticle dispersion bands are filled up to the Fermi energy, selected to be equal to zero for simplicity in the schematic Figs. 4, 2 and S1. If the wire is accelerated instantaneously to velocity $v$, the energetic escape condition for the highest-energy bound quasiparticles in vector form is $\mathbf{v}_{fl} \cdot \hat{\mathbf{p}}_{in} \geq \Delta/p_F + \mathbf{v} \cdot \hat{\mathbf{p}}_{out}$ where $\hat{\mathbf{p}}_{in}$ is a unit vector that corresponds to the bound quasiparticle's momentum during the increase in $\mathbf{v}_{fl}$ and $\hat{\mathbf{p}}_{out}$ is the direction of momentum for the quasiparticle escaping to bulk. The local flow velocity near the wire surface follows $\mathbf{v}_{fl} = 2v\cos(\theta)\hat{\theta}$, where $\hat{\theta}$ is the azimuthal unit vector perpendicular to the cylinder radius.

For increasing $v$, the escape condition is first satisfied for $\hat{\mathbf{p}}_{in} \uparrow\uparrow \mathbf{v}_{fl}$ and $\hat{\mathbf{p}}_{in} \uparrow\downarrow \hat{\mathbf{p}}_{out}$ for quasiparticles at the wire vertices ($\theta$ = 0) where $v_{fl}$ reaches its maximum. In this case, we get the well-known critical cross-bar velocity

$$v_c = \Delta/(3p_F) = v_L/3. \tag{2}$$

This process is schematically illustrated in Fig. 2. Note that a similar process acts on quasiholes, but they escape in the opposite direction because their momentum and velocity point in opposite directions. We can use similar arguments for deceleration from a steady state configuration at nonzero $v_{fl}$, obtaining the critical velocity $v'_c = \Delta/(2p_F) = v_L/2$.

At velocities significantly higher than $v_c$, the vector picture is more complicated. For simplicity, the main text and Figs. 2 and S1 only discuss scalar quantities along the direction of the external flow. At $v$ = 45 mm s⁻¹, the bulk escape process concerns about 90% of the crossbar surface (see Fig. 1b), but the largest contribution of the bound state escape originates from the vicinity of the vertices where $v_{fl}$ is the largest. Precise calculation of the distribution requires a three-dimensional treatment of the system, and such numerical simulations are left for the future.

We note that oscillatory motion has been speculated to overheat the bound state system enough to result in observable bound state escape below $v_L/3$,[61] but no such "pumping" is observed in our experiments. The pumping of quasiparticles towards higher energies is unlikely because the quasiparticle redistribution is governed by diffusion, as discussed below.

## Diffusion rate

Assuming there are no quasiparticle-quasiparticle collisions, we can estimate the mean free path in the surface layer as the longest distance that a quasiparticle can travel without changing the direction of the group velocity. This distance is $l_{\parallel} = 2\sqrt{2Ra\xi}$, where $R$ is the crossbar radius, $\xi$ is the coherence length, and $a\xi$ is the effective thickness of the surface layer (see Fig.1b). If we assume $a = 3$ as an estimate of the surface layer edge where the most energetic quasiparticles (the ones that the experiment is sensitive to) would be reflected[5], then $l_{\parallel} \approx 12\,\mu m$ at zero pressure. This yields the diffusion constant $D \sim l_{\parallel} v_{qp} = l_{\parallel} v_L$. Note that along the length of the crossbar cylinder, the system is homogeneous, and the diffusion experiment is not sensitive to the mean free path in this direction.

We can now solve the diffusion equation $\partial_t n = D\nabla^2 n$. Here $n$ stands for the part of the quasiparticle population that is out of equilibrium, $\nabla^2$ is the Laplace operator, and $\partial_t$ stands for the time derivative. The Fermi sea seen in Fig. S1 at $t = 0$ at energies below zero is uniformly filled, and there are no quasiparticles above zero energy. Thus, in that case, $n = 0$. The population carrying the momentum imbalance when the wire is stopped depends on the details of the dispersion relation, but we estimate that it initially follows the local flow velocity so that $n(\theta) \sim \cos\theta$. The resulting time dependence is $n(t,\theta) = n(t = 0,\theta)\exp(-tD/R^2)$, yielding the diffusion time constant $\tau = R^2/D$. Inserting $R = 63.5\,\mu m$ gives $\tau \approx 13$ ms. On the other hand, fitting $a$ to the experimentally observed time constant gives $a \approx 10$, which exceeds the theoretical expectation by a factor of ~2[5]. The feasibility of these values $a$ is discussed in the next section.

## Heat release from bound quasiparticles

The total energy released during the measurement cycle "up–up" (green + blue lines in Fig. 3) is $Q_{tot}^{up-up} = [Q + 2q] + [Q_{up} + q_{up} + q]$, where the brackets separate the contributions from the first and second phases of crossbar motion. Here $Q$ and $Q_{up}$ are due to the drag force from collisions with thermal bulk quasiparticles. The bulk thermal quasiparticle density increases by less than 10% in the course of a typical up cycle, thus $Q_{up} \approx Q$. Note also that $Q_{up}$ is independent of $\Delta t$, which means that the small difference between $Q$ and $Q_{up}$ can be ignored in the analysis of the bound state dynamics.

The first acceleration phase and each deceleration phase release approximately the same amount of heat originating from the bound quasiparticle system, denoted $q$. That is, the critical velocity for quasiparticle release from the acceleration is $v_L/3$ while that from the deceleration is $v'_c = v_L/2$. This means that the deceleration should release fewer quasiparticles than the acceleration, but the difference can be ignored if the wire moves at a speed much higher than $v_c$ (here $5v_c$). In the above expression for $Q_{tot}^{up-up}$ we have thus approximated that the heat released from the deceleration is equal to that from the acceleration.

The heat release from the second acceleration phase is $q_{up} = q(1 - \exp(-(\Delta t + t_0)/\tau))$. That is, the bound state population available for ejection during acceleration is depleted by the first cycle. Full depletion of the bound quasiparticles that are available for bulk escape implies $q_{up} = 0$, and after a long enough recovery time $q_{up} = q$. For intermediate values of $\Delta t$, the bound state population available for bulk ejection recovers exponentially owing to the diffusion with time constant $\tau$. Note that the bound state recovery starts as soon as the deceleration at the end of the first up cycle begins and continues until the same velocity is reached again during the following acceleration. This process is approximately accounted for by adding $t_0 = 6$ ms (sum of the acceleration and deceleration times) to the recovery time $\Delta t$ in the exponential decay expression.

For an "up–down" cycle (green+red lines in Fig. 3), the energy release is expected initially to be larger than for up-up cycles (see Fig. S1), decreasing as a function of $\Delta t$ and reaching the same level as the "up–up" cycles at large $\Delta t$. The time dependence is determined by the same diffusion process as detailed above. The total energy released is therefore $Q_{tot}^{up-down} = [Q + 2q] + [Q_{down} + \bar{q}_{down} + 2q]$. With similar arguments as above, $Q_{down} \approx Q$. Note, however, that $Q_{up} > Q_{down}$ owing to hysteresis in the distance covered by the crossbar[4]. This effect is proportional to the density of thermal bulk quasiparticles and therefore vanishes at the lowest temperatures measured in this Article.

The diffusion picture does not determine the magnitude of the time-dependent excess of bound quasiparticles ejected by the acceleration in the up-down cycle. Separating the asymptotic bound state contribution from the decaying excess, the excess heat release from the second phase of motion is denoted $\bar{q}_{down} = q_{down}\exp(-(\Delta t + t_0)/\tau)$. The number of bound quasiparticles released is a faster-than-linear function of velocity $v$ at $v > v_c$[4], which hints that $q_{down} > q$, as confirmed in the main text.

We can estimate the available quasiparticle energy release from the surface layer as follows. The gap suppression region around the crossbar has the volume $V = 2\pi Ra\xi L$ ($L$ is the crossbar length). Taking the diffusion calculation above literally yields the self-consistent thickness of the layer $a\xi$ with $a \approx 10$. The Doppler shift energy bridges the bulk escape at crossbar velocity $v = v_c$. If we assume the surface layer is populated according to the normal state density of states $N(0)$, then the energy release from the entire crossbar surface is $q \sim VN(0)\Delta^2(v/v_c - 1)^2 \approx 13$ pJ for $v = 45$ mm s$^{-1}$, in decent agreement with both the measured value with (12 pJ) and without (7 pJ) $^4$He pre-plating of the wire surface. Choosing a more conservative $a = 3$ yields $q \approx 4$ pJ. We emphasise that this estimate neglects many important contributions, such as the quasiparticle release from the legs of the moving superconducting wire (the inclusion of which would increase the energy) and the fact that a significant part of the crossbar surface has a smaller flow velocity than the maximum at the vertices, which corresponds to $v_c$ (thus, decreasing the released energy). Furthermore, according to ref. 61, page 296, the gap suppression near the wire is significantly expanded by the increased density of quasiparticles when $v > 2v_c$, which would act to increase the quasiparticle emission. This effect may explain why the fitted parameter $a$ appears so large.

## Coupling between bulk and surface superfluids

Once a number of bound quasiparticles have been emitted from the surface to the bulk superfluid, equilibrium is slowly recovered between the entire surface of the container and the bulk liquid. We have measured that in our experimental volume, the bulk superfluid temperature returns to the original level exponentially with a time constant of the order of $\tau_0 \sim 1$ s[54]. This process involves the entire surface area of the sample container, which is dominated by the silver sinter heat exchangers.

We can put a lower bound on the time constant that couples the bulk back to the surface. This process cannot be faster than the rate at which the bulk temperature recovers its original value. It can be slower, though, if the bulk primarily couples to the phonons in the heat exchanger sinters. If we scale the observed $\tau_0$ by the ratio of the surface areas of the goalpost wire crossbar and the entire sample container, this puts a lower bound on the direct re-equilibration onto the crossbar. The performance of the sinters in this sample container is analysed in ref. 54. It is not clear whether one should use the wall area covered by the sinter (called geometric sinter area in that reference) or the microscopic area of the sinters (which is orders of magnitude larger), but for a pessimistic estimate, we choose the smaller of the two, which is the geometric area (36 cm$^2$). This yields a lower limit of $\tau \sim 10^3$ s for the direct equilibration of the surface system on the crossbar via interaction with bulk quasiparticles.

This means thermal equilibrium between the bulk superfluid and the 2D surface system is likely re-established via the sinters. This is facilitated by the flow of quasiparticles along the legs of the goalpost wire. An experiment similar to ours but with a probe completely

detached from the walls, e.g. levitating sphere,[48,49] will allow measuring the equilibration time between 2D and 3D superfluids directly.

## Kapitza resistance and bound quasiparticle lifetime

The experimentally determined thermal Kapitza resistance between phonons in metal and superfluid $^3$He quasiparticles in the bulk superfluid is in the range $R_K \sim 10^4$ to $10^5$ Km$^2$ W$^{-1}$. Here the upper end of the range corresponds to a $^4$He preplated and the lower end to pure $^3$He interface between the fluid and the solid. These values are obtained by extrapolation to 1 mK using the data in ref. [62]. The temperature of the surface-bound quasiparticles is not above 1 mK because otherwise, they would escape to the bulk. For lower temperatures, the Kapitza resistance increases as $R_K \sim 1/T$[54]. For a pessimistic estimate of the decoupling of the crossbar phonons and the bound quasiparticles, we, therefore, take the measured pure $^3$He $R_K$ and assume the bound quasiparticle system is at 1 mK temperature during the decay described by $\tau$ in the main text.

The decay of energy via the Kapitza resistance is exponential with the time constant $\tau_{R_K} = R_K C$. Here $C$ is the heat capacity of the body of heat equilibrating via $R_K$. If we approximate the bound quasiparticle system to be a layer of normal fluid of thickness $a\xi$ at 1 mK right after the crossbar has stopped (consistent with the temperature chosen above), the resulting quasiparticle lifetime (decay rate of energy) in the bound quasiparticle system is $\tau_{R_K} \approx 12$ s for $a = 3$. Thus, even in a pessimistic estimate, exaggerating the heat flow, the bound quasiparticles exchange a negligible amount of energy with the wire phonons.

## Data availability

The data generated in this study have been deposited in the Lancaster University data repository at https://doi.org/10.17635/lancaster/researchdata/637[63].

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

## Acknowledgements
We thank Grigory Volovik, Jaakko Nissinen and Petri Heikkinen for stimulating discussions. This work was funded by UKRI EPSRC (EP/P024203/1: D.E.Z.; EP/X004597/1: V.T. and D.E.Z.; and EP/W015730/1: S.A.; EP/L000016/1: R.P.H., V.T. and G.R.P) and STFC (ST/T006773/1: R.P.H., V.T. and D.E.Z), as well as from the European Union's Horizon 2020 Research and Innovation Programme, under Grant Agreement no 824109 (European Microkelvin Platform: R.P.H., G.R.P., V.T. and D.E.Z). We acknowledge M.G. Ward and A. Stokes for their excellent technical support. S.A. acknowledges financial support from the Jenny and Antti Wihuri Foundation via the Council of Finnish Foundations.

## Author contributions
The measurements and data analysis were carried out by S.A., A.J., M.P., R.S., J.V., A.A.S. and D.E.Z. The experimental protocols were developed by R.S., J.V. and D.E.Z. Theoretical work, interpretation of the results, and background investigations were done by S.A. with contributions from R.P.H., G.R.P., V.T., V.V.Z. and D.E.Z. The manuscript was prepared by S.A. with help from G.R.P., while all authors provided comments and clarifications. D.E.Z. supervised the experiments.

## Competing interests
The authors declare no competing interests.
