## [Peer Review File · Nature Communications]

Transport of bound quasiparticle states in a two-dimensional boundary superfluidREVIEWER COMMENTS

Reviewer #1 (Remarks to the Author):

In this manuscript, Autti et al. studied the diffusion dynamics of quasiparticles bound near a boundary in superfluid $^3\text{He-B}$. By using a linearly moving wire, they studied the equilibrating process of bound quasiparticles returning back from the out-of-equilibrium distribution. They observed that the time constant and the heat released on the equilibrating process are consistent with those estimated from the diffusion process of quasiparticles within the boundary layer and concluded that quasiparticles bound near the boundary form an isolated two-dimensional Fermi system.

The paper itself is interesting. However, the experiment itself is almost the same as that reported previously by the author's group (Ref.1). In Ref.1, they found that there is a long time scale for the equilibrating process of bound quasiparticles, and they already concluded that quasiparticles bound near the boundary are independent of bulk quasiparticles. In my understanding, the major achievement of the present work is elucidating the fact that the diffusive motion within bound quasiparticles is responsible for the equilibrating process. Although the authors emphasize the two-dimensional nature of bound quasiparticles which is independent of bulk quasiparticles in this manuscript, the independence was already stated in Ref.1. So, I am skeptical about how much novelty presents in this study, whether this study clarifies a fundamental aspect of fermi superfluids, or whether the present paper attracts a broad audience. I think that this paper would find a more appropriate audience in a specialized journal.

I have some more remarks:

The authors use the term 'two-dimensional superfluid' in the title, abstract, and many places in the manuscript. I think that the use of this term is inappropriate and misleading. The system they studied is composed of three parts; bulk condensate of ^3He Cooper pairs, bulk quasiparticles, and quasiparticles trapped near the boundary. What the authors discuss in this manuscript is the dynamics of quasiparticles trapped near the boundary, which form a two-dimensional gas of quasiparticles. They are not two-dimensional superfluid. Furthermore, the authors use the term like 'two-dimensional condensate' (in the abstract, at line 49 of 1st page, etc.) The two-dimensional quasiparticles never condense in their experiments. These expressions are misleading to readers.

I found that this manuscript is difficult to read straightforwardly even for researchers working in the field of superfluid ^3He like me. When I read the manuscript for the first time, I had to move back and

forth and sometimes needed to read Ref.1 deeply. I think that it is more difficult for readers from other fields. The simplest way to improve the presentation is probably to move Fig. 7 and Fig. 8 and their descriptions before Fig. 2 or to combine Fig. 7, Fig. 8, and Fig. 2 in a single figure. Because the process of the heat release due to quasiparticle emission shown in Fig.8 is essential to understand this paper, it should be properly explained in the main text. In addition, it is difficult to read what is the major achievement of this paper.

In Fig.3, there is no description of what the pillars represent. It should be described in the caption.

I do not understand how the slope of the green line in Fig.4 is obtained, so I cannot figure out whether there is an agreement between the model used and the experiments. The slope of the green line is different from that of the black line. There is no description of how the green line is obtained.

In the conclusion written in lines 4-10, page 5, the authors describe that the surface provides a preferential path and that zero-temperature superfluid ^3He is thermo-mechanically two-dimensional. I feel that writing it in the conclusion is very odd because the authors did not study heat transport in this manuscript.

References:

line 28 on page 5: The following papers studied Majorana surface bound states formed at a free surface and are suitable to add;

Y. Tsutsumi, PRL 118, 145301 (2017), Scattering Theory on Surface Majorana Fermions by an Impurity in $^3\text{He-B}$.

H. Ikegami and K. Kono, J. Low Temp. Phys. 195, 343-257 (2019), Review: Observation of Majorana Bound States at a Free Surface of $^3\text{He-B}$.

line 31 on page 5: the following two papers are the first papers that pointed out the presence of Majorana surface states theoretically and should be added;

S.B. Chung and S.C. Zhang, PRL. 103, 235301 (2009), Detecting the Majorana Fermion Surface State of $^3\text{He-B}$ through Spin Relaxation.

A.P. Schnyder, S. Ryu, A. Furusaki, and A.W.W. Ludwig, Phys. Rev. B 78, 195125 (2008), Classification of topological insulators and superconductors in three spatial dimensions.

In addition, details of Majorana surface states are described in T Mizushima, et al., J. Phys. Soc. Jpn. 85, 022001 (2016), Symmetry-Protected Topological Superfluids and Superconductors -From the Basics to ^3He -.

Reviewer #2 (Remarks to the Author):

The manuscript reports interesting result and a novel interpretation of the slow relaxation of quasiparticles trapped near walls of objects submerged in superfluid helium-3 at ultra-low temperatures. At extremely temperatures the bulk of the fluid has no quasiparticles and experimentally observed relaxation is attributed to internal (diffusive) redistribution of quasiparticles within two-dimensional layer near the surface. The manuscript is clearly written and data and interpretations are convincing. I believe that the manuscript will be of general interest to a broad readership and am in favor of publication after several questions/comments listed below are addressed.

1. The authors provide a series of references to past work that have established the existence of surface bound state quasiparticles near the walls of objects placed into superfluid helium-3. However the authors have overlooked a few key works. To be more comprehensive they should include reference to the following: i) ii) A.B. Vorontsov and J. A. Sauls, Phys. Rev. B 68, 064508 (2003). ii) H. Choi et al., PRL 96, 125301 (2006). iii) J.P. Davis et al., PRL 101, 085301 (2008).

2. On the first page of the manuscript the authors refer to a "probe rod" placed in the superfluid. This term could strike the reader as confusing jargon. I suggest the authors to use "immersed mechanical probe" or something similar, which the have done later in the manuscript.

3. On second page, line 21, it would be helpful to the reader to understand how to obtain the $v_c = vL/3$ equation comes from for a cylindrical rod. Perhaps to include in the SI or at least a reference to prior work.

4. On second page, starting at line 79, the authors state that the density of bulk quasiparticles is "very low" and as such cannot be directly observed. The authors provide reference 1, which is helpful. But it would also be useful to state in the present manuscript quantitatively how long (even a bound) the expected time constant would be for bulk quasiparticle relaxation to compare with 6 ms.

5. On page 4, starting at line 60, the authors discuss the observation that despite the fact that their wire moves a "large bulk superfluid volume" at speeds exceeding the critical velocity they observe no generation of bulk quasiparticles. This does seem surprising and warrants some more discussion to understand. Or perhaps I not understand the authors meaning?

Responses to the referees for paper NCOMMS-23-11752-T.

We thank the Referees for their feedback. Our detailed response to all comments can be found below. Our response is in black and the original Referee comments are in blue.

In addition to the changes discussed below, we have also adjusted the format to comply with Nature Communications requirements. All the changes made are indicated in the redlined version of the manuscript that we have provided.

Reviewer #1 (Remarks to the Author):

In this manuscript, Autti et al. studied the diffusion dynamics of quasiparticles bound near a boundary in superfluid $^3\text{He-B}$. By using a linearly moving wire, they studied the equilibrating process of bound quasiparticles returning back from the out-of-equilibrium distribution. They observed that the time constant and the heat released on the equilibrating process are consistent with those estimated from the diffusion process of quasiparticles within the boundary layer and concluded that quasiparticles bound near the boundary form an isolated two-dimensional Fermi system.

The paper itself is interesting. However, the experiment itself is almost the same as that reported previously by the author's group (Ref.1). In Ref.1, they found that there is a long time scale for the equilibrating process of bound quasiparticles, and they already concluded that quasiparticles bound near the boundary are independent of bulk quasiparticles. In my understanding, the major achievement of the present work is elucidating the fact that the diffusive motion within bound quasiparticles is responsible for the equilibrating process. Although the authors emphasize the two-dimensional nature of bound quasiparticles which is independent of bulk quasiparticles in this manuscript, the independence was already stated in Ref.1. So, I am skeptical about how much novelty presents in this study, whether this study clarifies a fundamental aspect of fermi superfluids, or whether the present paper attracts a broad audience. I think that this paper would find a more appropriate audience in a specialized journal.

The main criticism of Referee #1 is that the present paper is similar to the previous work of Reference [1]. In the previous paper we described the experimental approach with a few hints as to what we thought might be contributing to the observed behaviour.

In the present paper we have made the big step forward in actually having observed, evidenced and analysed the dynamic motion of the bound state particles. We are not aware of prior experiments which probe any kind of transport in the surface system. There has been no theoretical consensus on whether the bound quasiparticles contribute to macroscopic transport in this system. Therefore, our work represents a major advance in the understanding of Fermi superfluids.

That said, it is our responsibility to make this message clear and we have reworded the title and the first paragraph and several other points in the text to emphasize this point.

I have some more remarks:

The authors use the term ‘two-dimensional superfluid in the title, abstract, and many places in the manuscript. I think that the use of this term is inappropriate and misleading. The system they studied is composed of three parts; bulk condensate of ^3He Cooper pairs, bulk quasiparticles, and quasiparticles trapped near the boundary. What the authors discuss in this manuscript is the dynamics of quasiparticles trapped near the boundary, which form a two-dimensional gas of quasiparticles. They are not two-dimensional superfluid. Furthermore, the authors use the term like ‘two-dimensional condensate’ (in the abstract, at line 49 of 1st page, etc.) The two-dimensional quasiparticles never condense in their experiments. These expressions are misleading to readers.

The components of the B phase order parameter amplitude (superfluid gap) are uniform in all momentum directions, and thus thermal quasiparticles (normal fluid) vanish exponentially as temperature decreases to zero. In the superfluid A phase, the superfluid gap is zero in one momentum direction and in the superfluid polar phase in two momentum directions. Thus, in these phases the thermal excitation density does not go to zero exponentially but following a power law instead. These systems are superfluids despite the power-law temperature dependence, and they are separate superfluid phases from the B phase as implied by their different order parameter structures. It is common to describe, say, ^4He below the lambda transition as a superfluid. This comprises a condensate component and a “normal” component of excitations.

Let us apply these observations to the B phase region within a coherence length of container walls. The surface region is a superfluid because the gap is not uniformly zero. The gap spectrum is different from the bulk phases, so it is a separate superfluid phase. As agreed by the referee, this system is two-dimensional as measured by the characteristic length scale in the system. Thus, it is a two-dimensional superfluid. The quasiparticles do condense with a power-law temperature dependence, like in the bulk A phase. Thus, describing the system as a condensate is justified.

We would also point out that the observed transport time constant is determined by the bound state group velocity which arises from the gap not being uniformly zero, and that the estimated bound state release to the bulk is consistent with the increased density of quasiparticles in the surface layer, arising from the suppressed gap spectrum.

We have completely rewritten the relevant parts of the paper, adding three paragraphs in the concluding section of the main text to explain in detail why the surface layer is a separate superfluid system and modifying the abstract to make this perspective clear. We have also tightened up the language elsewhere, where relevant.

I found that this manuscript is difficult to read straightforwardly even for researchers working in the field of superfluid ^3He like me. When I read the manuscript for the first time, I had to move back and forth and sometimes needed to read Ref.1 deeply. I think that it is more difficult for readers from other fields. The simplest way to improve the presentation is probably to move Fig. 7 and Fig. 8 and their descriptions before Fig. 2 or to combine Fig. 7, Fig. 8, and Fig. 2 in a single figure. Because the process of the heat release due to quasiparticle emission shown in Fig.8 is essential to understand this paper, it should be properly explained in the main text. In addition, it is difficult to read what is the major achievement of this paper.

The emission of bound quasiparticles to the bulk superfluid is a technique used in this paper to study the dynamics of the bound quasiparticles that remain bound to the surface. As noted by the Referee above, this technique is reported in our earlier papers, in particular Ref. [1]. Details on the use of this technique, in our view, belong in Methods. As a compromise, we have moved Figure 7 before original Figure 2 but left original Figure 8 in Methods. We have also improved the introduction to make the paper easier to read. The abstract, introduction and conclusions have been rewritten to make the main achievements that we are dealing with the dynamics of the bound excitations more prominent.

In Fig.3, there is no description of what the pillars represent. It should be described in the caption.

The pillars are indeed described in the caption: "...The vertical bands of colour indicate where surface quasiparticles are emitted from the wire during acceleration and deceleration....". However, this might be more clearly written. We have amended the wording.

I do not understand how the slope of the green line in Fig.4 is obtained, so I cannot figure out whether there is an agreement between the model used and the experiments. The slope of the green line is different from that of the black line. There is no description of how the green line is obtained.

The green line is a linear fit to the data. This is now explicitly stated in the caption. The black line is an estimate detailed in Methods. We have rearranged and rewritten the subsection in Methods that details this estimate. This explanation is now also separated in its own subsection for clarity. Note that the estimate is obtained by extrapolating the measured Andreev reflection velocity dependence to a velocity much beyond the critical velocity (45mm/s vs. 9mm/s) and thus the agreement with the data is good. This is also now explained in the caption.

In the conclusion written in lines 4-10, page 5, the authors describe that the surface provides a preferential path and that zero-temperature superfluid ^3He is thermo-mechanically two-dimensional. I feel that writing it in the conclusion is very odd because the authors did not study heat transport in this manuscript.

In a Cooper-pair superfluid, quasiparticles represent the entire thermal bath. Thus, transport experiments on quasiparticles are equivalent to heat transport experiments. We show that the energy carried by the bound quasiparticles is not absorbed by the container wall (wire crossbar) and does not leak to the bulk superfluid. The latter aspect has been clarified in this revised version of the manuscript as detailed in our response to Referee B below. This means that the surface forms a preferential path for heat flow. We have reformulated the paragraph to make this clear. The quasiparticle transport mechanism that we are studying in this paper may also explain the results of anomalous thermal conductivity observed in [27], which we now emphasise in the conclusions.

References:

line 28 on page 5: The following papers studied Majorana surface bound states formed at a free surface and are suitable to add;

Y. Tsutsumi, PRL 118, 145301 (2017), Scattering Theory on Surface Majorana Fermions by an Impurity in $^3\text{He-B}$.

H. Ikegami and K. Kono, J. Low Temp. Phys. 195, 343-257 (2019), Review: Observation of Majorana Bound States at a Free Surface of $^3\text{He-B}$.

line 31 on page 5: the following two papers are the first papers that pointed out the presence of Majorana surface states theoretically and should be added;

S.B. Chung and S.C. Zhang, PRL. 103, 235301 (2009), Detecting the Majorana Fermion Surface State of $^3\text{He-B}$ through Spin Relaxation.

A.P. Schnyder, S. Ryu, A. Furusaki, and A.W.W. Ludwig, Phys. Rev. B 78, 195125 (2008), Classification of topological insulators and superconductors in three spatial dimensions.

In addition, details of Majorana surface states are described in T Mizushima, et al., J. Phys. Soc. Jpn. 85, 022001 (2016), Symmetry-Protected Topological Superfluids and Superconductors -From the Basics to $^3\text{He-}$.

These references have been added as suggested. We have also added three early papers on the suppression of the order parameter near walls (references [1-3] in the revised manuscript).

Reviewer #2 (Remarks to the Author):

The manuscript reports interesting result and a novel interpretation of the slow relaxation of quasiparticles trapped near walls of objects submerged in superfluid helium-3 at ultra-low temperatures. At extremely temperatures the bulk of the fluid has no quasiparticles and experimentally observed relaxation is attributed to internal (diffusive) redistribution of quasiparticles within two-dimensional layer near the surface. The manuscript is clearly written and data and interpretations are convincing. I believe that the manuscript will be of general interest to a broad readership and am in favor of publication after several questions/comments listed below are addressed.

We thank the Referee for these remarks and for the constructive suggestions below. These have resulted in a much-improved manuscript.

1. The authors provide a series of references to past work that have established the existence of surface bound state quasiparticles near the walls of objects placed into superfluid helium-3. However the authors have overlooked a few key works. To be more comprehensive they should include reference to the following: i) ii) A.B. Vorontsov and J. A. Sauls, Phys. Rev. B 68, 064508 (2003). ii) H. Choi et al., PRL 96, 125301 (2006). iii) J.P. Davis et al., PRL 101, 085301 (2008).

We have added these references as suggested. Phys. Rev. B 68, 064508 (2003), which concentrates on the A phase of ^3He , is referenced on page three where the 2D transport picture is constructed. The other two references are included on page one where the prior evidence for B phase surface physics is discussed.

2. On the first page of the manuscript the authors refer to a "probe rod" placed in the superfluid. This term could strike the reader as confusing jargon. I suggest the authors to use "immersed mechanical probe" or something similar, which they have done later in the manuscript.

We have reformulated this sentence as suggested.

3. On second page, line 21, it would be helpful to the reader to understand how to obtain the $v_c = v_L/3$ equation comes from for a cylindrical rod. Perhaps to include in the SI or at least a reference to prior work.

We have reformulated that paragraph to contain an explanation of the critical velocity for cylindrical probes. We have also added a reference to Methods where the critical velocity is derived more rigorously and renamed the relevant section in Methods to emphasise this. Note also that a technical mistake in the derivation in Methods in the description of the quasiparticle ejection process has been removed.

4. On second page, starting at line 79, the authors state that the density of bulk quasiparticles is "very low" and as such cannot be directly observed. The authors provide reference 1, which is helpful. But it would also be useful to state in the present manuscript quantitatively how long (even a bound) the expected time constant would be for bulk quasiparticle relaxation to compare with 6 ms.

We have added an estimate of the thermalisation time between the crossbar surface and the bulk based on the thermalisation of the bulk via the silver sinters. We have also added a paragraph in Methods that details how we arrived at this estimate.

5. On page 4, starting at line 60, the authors discuss the observation that despite the fact that their wire moves a "large bulk superfluid volume" at speeds exceeding the critical velocity they observe no generation of bulk quasiparticles. This does seem surprising and warrants some more discussion to understand. Or perhaps I not understand the authors meaning?

We have expanded this discussion. This comment is referring back to an experiment a few years ago where it was demonstrated that moving a massive object through superfluid ^3He at above the Landau critical velocity does not lead to dissipation and pair breaking as the surface layer of superfluid in a sense shields the bulk superfluid from the motion of the object. (See: "Breaking the superfluid speed limit in a fermionic condensate", Nature Physics 12, 1017 (2016)). This was a completely unexpected at the time but has since led to several papers discussing this effect.

First, we have added a reference to a recent theory paper discussing the expectation of such pair breaking in the bulk. Second, we show that our data rules out such mechanism by a large margin. We have also added a speculative reason why such direct pair breaking perhaps does not occur.

REVIEWERS' COMMENTS

Reviewer #1 (Remarks to the Author):

The authors improved the manuscript based on the reviewers' comments, and now its quality is better. However, I am still skeptical that this study has clarified an important aspect of Fermi superfluid. In particular, I cannot figure out at what point the understanding of the transport of bound quasiparticles contributes to a major advance in the fundamental understanding of the Fermi superfluid. In their reply, the authors described that there has been no theoretical consensus on whether the bound quasiparticles contribute to macroscopic transport. However, this is not a justification for the importance of this study. I still think that this paper is difficult to attract a broad audience, and I do not recommend it for publication.

Reviewer #2 (Remarks to the Author):

In their revised manuscript the authors have thoroughly addressed all the points raised by the reviewers in the initial round of review. In my opinion the manuscript is now suitable for publication in Nature Communications.

Responses to the referees for paper NCOMMS-23-11752A.

Our response to all Referee comments is in black and the original Referee comments are in blue.

Please note that the technical Figure 8 has been updated so to provide the thermometry in experimental units, following the convention applied in the rest of the paper (thermometer resonance width). The caption contains an approximate conversion to a change in temperature. Figure 7 has been replaced by a re-rendered version to remove licence overlap with previously published material.

We have also provided a redlined version of the manuscript that indicates the changes made.

Reviewer #1 (Remarks to the Author):

The authors improved the manuscript based on the reviewers' comments, and now its quality is better. However, I am still skeptical that this study has clarified an important aspect of Fermi superfluid. In particular, I cannot figure out at what point the understanding of the transport of bound quasiparticles contributes to a major advance in the fundamental understanding of the Fermi superfluid. In their reply, the authors described that there has been no theoretical consensus on whether the bound quasiparticles contribute to macroscopic transport. However, this is not a justification for the importance of this study.

If superfluid $^3\text{He-B}$, a prototype Fermi superfluid, is shown to be made of two superfluids, the well-known bulk and the novel 2D surface superfluid, it is difficult to see how this is not an important advance in fundamental understanding of the system. Our transport experiments show that the surface system is a well-defined, independent superfluid. We are not aware of any prior experimental evidence (nor has the Referee pointed out any) that would substantiate such a conclusion.

I still think that this paper is difficult to attract a broad audience, and I do not recommend it for publication.

We will pay particular attention to accessibility and attractivity to a broad audience in the remaining aspects of the publication process, such as the editorial summary of the work that will appear on the Nature Communications front page.

Reviewer #2 (Remarks to the Author):

In their revised manuscript the authors have thoroughly addressed all the points raised by the reviewers in the initial round of review. In my opinion the manuscript is now suitable for publication in Nature Communications.

We thank the Referee for the valuable feedback.